# Investigating the biochemical variations in onion leaves due to purple blotch disease and its management through induced resistance

Muhammad Asghar[1], Muhammad Atiq[1], Muhammad Usman Ali[1], Ghalib Ayaz Kachelo[1,2], Nasir Ahmad Khan[1], Khalid Naveed[3], Muhammad Usman[1], Ahmad Nawaz[1], Owais Iqbal[4], Nasir Ahmed Rajput[1]*

1 Department of Plant Pathology, University of Agriculture Faisalabad, Faisalabad, Pakistan, 2 Crop Disease Research Institute, Southern Zone Agricultural Research Center, PARC, Karachi, Pakistan, 3 Department of Plant Pathology, Sub Campus Depalpur, Okara, University of Agriculture Faisalabad, Faisalabad, Pakistan, 4 State Key Laboratory for Conservation and Utilization of Bio-Resources in Yunnan, Yunnan Agricultural University, Kunming, Yunnan, China

* nasirrajput81@gmail.com, nasir.ahmed@uaf.edu.pk

## Abstract

Purple blotch (PB), caused by *Alternaria porri* (Ellis) *ciferri*, poses a significant threat to onion crop, resulting in major economic losses in both bulb and seed production globally. The incidence of this disease underscores the critical need for an effective management. In the present study, screening of onion genotypes for PB under field conditions revealed that the genotypes like Phulkara and Ceylon expressed resistant, response with 8.21 and 8.91% disease severity index (DSI) respectively, whereas Desi Black and Red Imposta were highly susceptible, with DSI of 67.38 - 79.41%. Furthermore, resistant and susceptible genotypes were also evaluated for biochemical variation analysis. Significant variations ($p \leq 0.05$) in antioxidant enzymes were observed across reaction groups (inoculated and un-inoculated), types (resistant and susceptible), and onion varieties in response to *A. porri* infection. The analysis of variance showed significant changes in antioxidant enzymes level of onion leaves, including catalase (CAT), peroxidase (POD), and super dismutase (SOD). Results indicated that the concentration of SOD (55.37%), POD (41.87%) and CAT (37.92), respectively, in resistant plant leaves, whereas susceptible varieties showed SOD (43.71%), POD (28.3%) and CAT (26.59%). Furthermore, the amount of antioxidant enzymes was reduced in both resistant as well as susceptible varieties of onion after inoculation. Amount of SOD (68.04%), POD (57.45%) and CAT (50.87%) were recorded in un-inoculated group of onion plants that reduced to 31.05, 12.98 and 13.64% in inoculated group respectively. For the management of *A. porri* four different plant activators (salicylic acid, benzoic acid, citric acid and di-potassium hydrogen phosphate) at three different concentrations (0.5, 0.75 and 1%) were evaluated under greenhouse and field conditions. Among all salicylic acid was found to be

**Data availability statement:** All relevant data are within the manuscript and its Supporting Information files

**Funding:** The author(s) received no specific funding for this work.

**Competing interests:** The authors have declared that no competing interests exist.

most effective in controlling this disease under greenhouse and field conditions. The present study revealed that purple blotch affects the biochemical mechanism of the plants, which helps in activating the resistance process against pathogens through various enzymes. Additionally, salicylic acid demonstrated significant efficacy in controlling purple blotch in onions.

## Introduction

The onion (*Allium cepa* L.) belongs to the Amaryllidaceae family and is a short-duration, self-pollinated crop grown in regions ranging from temperate to semi-arid. It is the second most commonly produced and consumed vegetable worldwide, valued for their economic, nutritional, and medicinal benefits [1]. The Total area of the world under onion cultivation is 2.7 million hectares, with a production of 46.7 million tons [2]. In Pakistan, onion crop is planted on an area of 135.49 thousand hectares, producing 1.85 million tons [3]. Onion is a rich source of several minerals, including zinc, potassium, and vitamin E, containing beneficial compounds like quercetin, fructans, and organosulfur compounds, which help reduce the risk of diseases such as cancer, osteoporosis, and possess various health properties including anti-inflammatory, antibacterial, and analgesic effects. Onion by-products, such as skins, peels, stalks, and residual materials generated during onion processing or consumption, are often used for various purposes, such as animal feed, composting, or extraction of bioactive compounds [4].

PB is a serious disease, caused by *Alternaria porri,* leads to substantial economic losses in onion crops, while specific global loss data is limited, but in our neighboring countries like India, yield losses has been reported in both seed and bulb crops, ranging from 2.5 to 97percent [5]. The symptoms on onion leaves caused by this pathogen appear as small white sunken patches. These patches expand, become zonate, and develop a purple color, leading to bulb tissue rot and yellowing at the necks [6]. The pathogen harms the leaf tissue by utilizing its nutrients as an energy source, disrupting the process of bulb initiation and leading to delayed bulb formation and maturation [7]. Specific environmental factors, such as a high relative humidity about 80–90% and a moderate temperature of 25–30°C, favor the development of this disease [8]. The influence of PB on onion highlights the importance of implementing effective management strategies, including the use of synthetic fungicides, resistant cultivars, proper sanitation practices, biological controls such as beneficial microorganisms, nutritional controls like balanced fertilization, and the application of phyto-extracts.

The pathogen affects the activity of key enzymes such as peroxidase (POD), catalase (CAT), and superoxide dismutase (SOD) in diseased plants [9]. The presence of malondialdehyde (MDA) serves as an indicator of oxidative damage caused by various stresses [10]. During the host-pathogen interaction,an increase in reactive oxygen species (ROS) like hydrogen peroxide ($H_2O_2$), superoxide ($O2^-$), and hydroxyl radicals ($OH^-$). This elevation in ROS level raises MDA concentration, which in turn induces fatty acid peroxidation, resulting in membrane damage [11].

These biochemical changes play a crucial role in disease progression and pathogen invasion, providing valuable insights for developing robust disease management strategies [12]. The shift from symptom-based identification to biochemical and genetic methods is driven by the limitation of symptom reliability and the genetic heterogeneity among pathogen [13]. Monitoring biochemical shifts can help to predict plant tolerance and susceptibility, reflecting the presence and pattern of disease and pathogens. These biochemical markers are essential for assessing genetic diversity and developing an effective disease control measure, enabling early-stage interventions and improving overall disease management approaches [14]. Cultural methods offer limited effectiveness in managing PB, so, long-term strategy should focus on resistant genetic resources containing resistance genes. The use of resistant germplasm becomes important during epidemics, making germplasm screening essential for both economic and ecological sustainability. Utilizing resistant seeds or plant material provides a simple and cost-effective approach to disease management [15].

Chemical fungicides and pesticides pose an environmental risk due to their residual effects. The challenges of pathogen resistance and environmental concerns are driving the search for sustainable alternatives. One promising area is the development of plant defenses, including systematic acquired resistance (SAR). SAR activates plant defense mechanisms by through e salicylic acid, benzoic acid, offering potential for effective disease control [16]. Salicylic acid and Jasmonic acid (SA-JA crosstalk) play crucial roles in coordinating plant immunity, offering vital protection against pathogens [17]. Potassium salts, like $K_2HPO_4$ and potassium silicate, have been effective in managing purple blotch in onion seed production by reducing disease severity and enhancing phenols and sugars level [18]. Additionally, potassium citrate, a significant organic acid involved in plant respiration, acts as a non-enzymatic antioxidant, helping to extend cell shelf life and promoting growth [19]. The objectives of this study were to evaluate the varietal response of onions against purple blotch (PB) under natural field conditions and to assess the biochemical alterations in inoculated and un-inoculated onion leaves due to the purple blotch fungus (A. porri). Additionally, the study aims to evaluate the effectiveness of plant activators against purple blotch of onion under both greenhouse and field conditions.

## Materials and methods

### Survey and sample collection

A comprehensive survey was conducted in different onion growing areas of District Faisalabad. Diseased samples having typical symptoms of purple blotch were collected at Chak 35 JB Parvinabad region (31°25'30.5"N 72°54'12.8"E), Faisalabad, Punjab, Pakistan. All bags were labeled with date, name, and place of sample collection. The samples were brought in Phytopathology laboratory, Department of Plant Pathology, University of Agriculture, Faisalabad and processed for isolation of associated pathogen.

### Isolation, purification and identification of *Alternaria porri*

The infected leaf samples were washed with tap water and cut down into smaller pieces (4–5 mm) with the help of sterilized scissor. These sections were surface sterilized with 75% ethyl alcohol for 10 seconds, and washed twice using distilled water. The specimens were dried with sterilized tissue paper. PDA media was poured into Petriplates and allowed to solidify. Using sterilized forceps, the disease samples were placed one by one onto the PDA media in the Petri plates, which were wrapped with paraffin tape to prevent contamination. All procedure was conducted in a laminar flow chamber. The Petri plates were incubator at 25±5 °C, and fungal growth was monitored regularly [20].

The mycelium from the advancing edges of developing colonies on the agar plates was carefully isolated to obtain pure cultures. The fungal pieces were transferred to new sterilized petri plates containing fresh potato dextrose agar (PDA) and incubated at 25°±5°C for seven days. The cultural characteristics of the fungus isolated from the infected onion leaves were thoroughly examined and tentatively identified using standard protocols. Only pure cultures of A. porri were used for laboratory and field trials, ensuring that no impurities were present [21]. The pathogen was identified through the examination of its morphological characteristics of the isolated pathogen.

## Proving the Pathogenicity

The pathogenicity test was performed by following Koch's postulates. For this purpose, an onion nursery collected from Ayub Agriculture Research Institute (AARI) Faisalabad was planted in earthen pots containing sterilized soil. After 20 days of nursery transplantation, healthy onion leaves were inoculated by two methods: placing a small amount of mycelium (10 mm Disc) from one-week-old culture of *A. porri* and spraying a spore suspension. The inoculated plants were regularly observed for the appearance of the disease symptoms. To fulfill Koch's postulates, the pathogen was re-isolated from the diseased leaves to confirm its association with the disease [22].

## Experimental setup for screening of onion genotypes against purple blotch

Eight weeks old seedlings of ten different genotypes of onion (Phulkara, Ceylon, Early red, Desi red, Mirpurkhas, Sultan F1, VRIO-9, Desi black and Red Imposta) were transplanted in field at Vegetables' Research Area, Ayub Agriculture Research Institute (AARI), Faisalabad located at longitude 73º74 East, latitude 30º31.5 North at an altitude of 184m above from sea level in Pakistan. These plants were transplanted on 75 cm wide raised beds by maintaining row-to-row (R-R) distances of 35 cm and plant-to-plant (P-P) distances of 8 cm under a Randomized Complete Block Design (RCBD), with three replications of each variety, and each replication contained three rows with a total of 10 plants. All agronomic practices (irrigation and fertilizers) were applied on proper time to ensure healthy plant growth. Fifteen days after transplantation, plants were artificially inoculated with a spore suspension @ $1 \times 10^6$ spores/mL using a hand sprayer in the evening, while control was maintained by applying distilled water. Irrigation was done after inoculation to provide maximum humidity for infection. Moreover, a susceptible variety as spreader was transplanted around the experiment to provide maximum inoculum pressure. For disease scoring, five plants from each replication were randomly selected and tagged to track disease progression consistently and minimize variability among individual plants during subsequent recordings. The disease severity index was calculated after seven days interval after inoculation till maximum disease was appeared (5 weeks after inoculation). Individual plant scoring was done by disease rating scale presented by Sharma [23] (Table 1).

Following individual plant scoring, disease severity index was calculated on percentage basis by following a formula given by Chiang [24]. On the basis of DSI value, all tested genotypes were categorized into different categories viz. Highly susceptible with more than 60% DSI (HS), susceptible with 40–60% DSI (S), Moderately susceptible with 20–40% DSI (MS), moderately resistant with 10–20% DSI (MR), resistant with 5–10% DSI (R) and immune with less than 5% DSI (I) as suggested by Pathak [25].

$$DSI\ (\%) = \frac{Sum\ of\ Class\ Frequency \times Score\ of\ Rating\ Class}{Total\ No.\ of\ Plants\ \times Maximal\ Disease\ Severity} \times 100$$

**Table 1. Disease rating scale used for screening of onion germplasm against purple blotch.**

| Percent Disease Incidence (PDI) | Scale (0–5) | Reaction |
|---|---|---|
| <5 | 0 | Immune |
| 5-10 | 1 | Resistant |
| 11-20 | 2 | Moderately resistant |
| 21-40 | 3 | Moderately susceptible |
| 41-60 | 4 | Susceptible |
| >60 | 5 | Highly susceptible |

## Estimation of biochemical variations in healthy and diseased onion leaves in response to purple blotch disease caused by *Alternaria porri*

**Collection of diseased samples.** Healthy and diseased leaves from two resistant (Phulkara and Ceylon) and two susceptible (VRIO-9 and FSD Red) onion varieties, based on screening results, were collected in separate brown bags from the Vegetable Research Area of Ayub Agriculture Research Institute (AARI), Faisalabad. These samples were brought to the Plant pathogen interaction laboratory, Department of Plant Pathology, University of Agriculture, Faisalabad.

**Preparation of samples for biochemical analysis.** The collected samples were subjected to two washes: first with tap water and then with distilled water. Approximately 10 mg of leaves were weighed and uniformly cut into small pieces for grinding. The leaves were then ground using a pestle and mortar with an extraction buffer containing 5 mL of $Na_2HPO_4$ and 5 mL of $NaH_2PO_4$. The resulting mixture was transferred to Eppendorf tubes and centrifuged at 10,000 rpm for 10 minutes using a "Table Top Centrifuge". The supernatant was filtered, transferred to sterilized bottles, and stored in a refrigerator set at 4°C (model PEL, PRGD-145).

**Superoxide dismutase (SOD) activity.** SOD activity was determined based on its ability to inhibit the photo-reduction of Nitrobluetetrazolium (NBT). To determine this reaction solution (3 ml) were prepared from both diseased and healthy leaf samples. The solution was mixed in separate test tubes, each containing 50 mM phosphate buffer (pH 7.8), 50 µl NBT, 100 µl methionine, 100 µl Triton, and 100 µl of riboflavin. The test tubes were then subjected to UV light for 15 minutes under a fume hood. The absorbance was measured using a spectrophotometer at 560 nm (BEL: Model 1.24). This was done by loading an ELISA plate (Ultra Cruz ELISA Plate with 96 wells) with 50 µl of leaf extract, 1 ml $H_2O_2$, and 100 µl of the reaction mixture. The amount of enzyme restricted 50% of NBT photo-reduction was considered as one unit of SOD [26].

**Peroxidase dismutase (POD) activity.** A reaction solution was prepared using both diseased and healthy leaf samples. The solution comprised of 50 µL of phosphate buffer (pH 7), 375 µL of $C_7H_8O_2$, 100 µL of $H_2O_2$, and 50 µL of leaf extract. Samples of 100 µL were then loaded onto a micro-plate (Ultra Cruz ELISA Plate, 96 wells), and changes in the absorbance of the reaction solution were recorded at 470 nm using a spectrophotometer (BEL: Model 1.24). One unit of catalase and peroxidase activity was defined as an absorbance change of 0.01 units per minute [26].

**Activity of Catalase (CAT).** Following the method discussed by [26], the CAT activity in both diseased and healthy leaves was determined using a reaction solution. This solution was prepared by combining 50µl of phosphate buffer (pH 8.3), 50µl of $H_2O$, and 100µl of enzyme extract. A total of 100 µl of the reaction mixture was loaded onto an ELISA micro-plate (Ultra Cruz ELISA Plate, 96 wells). Absorbance measurements were taken at 400 nm by using a spectrophotometer (BEL: Model L.24). One unit of CAT activity was defined as an absorbance change of 0.01 units per minute.

## Evaluation of plant activators against purple blotch of onion caused by *Alternaria porri*

**Greenhouse experiment.** A pot experiment was conducted in a greenhouse at research area of Department of Plant Pathology, UAF. In this experiment, 45-days-old onion seedlings were transplanted in earthen pots (17 × 15 cm) containing sandy loam soil. After 20 days of nursery transplantation, the plants were artificially inoculated using a fungal spore suspension. On the second day post-inoculation, four plant defense activators (Salicylic acid, Benzoic acid, Citric-acid and Di-Potassium hydrogen phosphate) were applied in three different concentrations (0.5, 0.75 and 1%), which were prepared by dissolving 0.5, 0.75 and 1g of each plant activator in 100 mL of distilled water. These solutions were applied in spray form using a manual hand sprayer. Three applications of the plant defense activators were made at seven-days interval. The experiment was designed using a Completely Randomized Design (CRD) with three replications of each treatment and a control. Disease incidence data was recorded three times, on the 7th, 14th and 21st days after the final application of the treatments.

**Field experiment.** The most effective concentration (1%) of each plant activator, identified under greenhouse conditions, was further tested in open field conditions. The nursery of a susceptible onion variety was collected from the Vegetable Research Institute, AARI, and transplanted onto 75 cm wide raised beds, maintaining row-to-row (R-R) spacing

of 35 cm and plant-to-plant (P-P) spacing of 8 cm at the research area of the Department of Plant pathology, UAF. After three weeks of nursery transplantation, the plants were inoculated with a spore suspension. Subsequently, plant defense activators (salicylic acid, benzoic acid, citric-acid and di-potassium hydrogen phosphate) were applied by spraying three times at one-week intervals using a hand sprayer, while control plants were treated with distilled water.

The field trial was arranged using a Randomized Complete Block Design (RCBD) with three replications for each treatment. Disease incidence data (%) was recoded for three consecutive weeks, starting seven days after the third application of the plant activators using the formula given by Kumar [27].

$$Disease\ incidence\ =\ \frac{Number\ of\ infected\ plants}{Number\ of\ total\ plants} x\ 100$$

## Statistical Analysis

Analysis of variance (ANOVA) was used to test the effects of different treatments on disease incidence and biochemical attributes of on onion plants with the least significant difference (LSD) at $p < 0.05$ using Statistics 8.1 software. Nested Structured Design was used to analyzed biochemical variation in inoculated un-inoculated and un-inoculated leaves of onion. Figures of recorded data were made by using OriginPro 2023.

## Results

### Isolation and Identification of pathogen *'Alternaria porri'*

The pathogen was isolated from the onion samples showing purple blotch symptoms. Initially, the growth of the fungus (*A. porri*) appeared as circular to irregular, flat or slightly raised colonies with a white to pale gray color. As the culture matured, it typically turned dark gray or olive-green, often exhibiting a velvety texture (Fig 1a).

The fungal pathogen was identified based on morphological characteristics and their comparison with the available literature. During the examination, the fungus showed septate mycelium having conidiophores emerging singly or in groups. Conidiophores were seen to be straight or flexible, occasionally geniculate, septate, and varied in colors from light to mid-brown (Fig 1b).

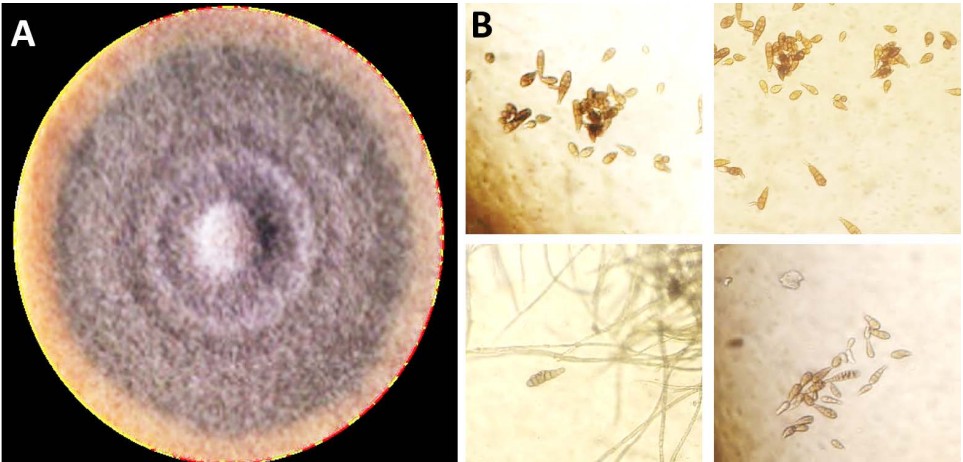

**Fig 1.  (a) The purified culture of *Alternaria porri*, isolated from the diseased leaves of onion showing typical symptoms of purple blotch (b) the microscopic picture of spores of *A. porri*.**

## Pathogenicity Assay

Pathogenicity test was performed to fulfil Koch's postulates. On the leaves of the artificially inoculated plants, typical purplish zonate marks were observed (Fig 2). The pathogen was re-isolated from the diseased leaves, and the purified growth was compared with the mother culture of the fungus based on morphological and microscopic characters. All the morphological and microscopic characters of re-isolated pathogen were closely matched those of the initial pure mother culture of *Alternaria porri* (Ellis) cif. and confirming it as the disease-causing agent for purple blotch in onion.

## Onion varietal response against purple blotch (PB) under natural field conditions

During field experiments, all varieties were affected by the disease, but the severity varied. Among them, Desi Black and Red Imposta were the most vulnerable, with high disease levels of 67.38% and 79.41%, earning them a rating of 5 on the highly susceptibility scale. On the other hand, VRIO-9 and Faisalabad Red showed susceptibility but to a lesser extent, with disease incidences of 44.53% and 54.45%, and were rated 4. Mirpurkhas and Sultan F1 displayed moderate susceptibility, with disease severity levels of 23.33% and 35.59%, giving them a rating of 3. In contrast, Early Red and Desi Red showed some resistance, with lower disease levels of 13.73% and 17.13%, earning a rating of 2. The most resistant varieties were Phulkara and Ceylon, with 8.21% and 8.91% disease incidence, securing a rating of 1. However, none of the tested varieties were completely immune to purple blotch disease. (Table 2).

## Biochemical alterations in inoculated and un-inoculated leaves of onion due to purple blotch fungus *(A. porri)*

**Estimation of SOD (U/mg protein).** Nested structured ANOVA was employed to determine the superoxide dismutase (SOD U/mg protein) activity in inoculated and un-inoculated onion leaves. The statistical analysis expressed that the group (susceptible and resistant) contributed 8.83% and the varieties 1.23% of the total variance, both showing significant difference in SOD concentration (Table 3). The maximum percentage of SOD (58.16 U/mg protein) was recorded in the resistant advanced line Ceylon, followed by Phulkara (52.58 U/mg protein). Conversely, the minimum concentration (42.54 U/mg protein) was recorded in the susceptible advanced line VRIO-9, followed by FSD Red(44.89 U/mg protein).A substantial difference in SOD concentration was recoded between inoculated (31.05 U/mg protein) and un-inoculated (68.04 U/mg protein) plants, according for 88.21% of the total variance. Additionally, a significant difference was found between resistant (55.37 U/mg protein) and susceptible (43.71 U/mg protein) onion plants (Table 3 and 4; Fig 3).

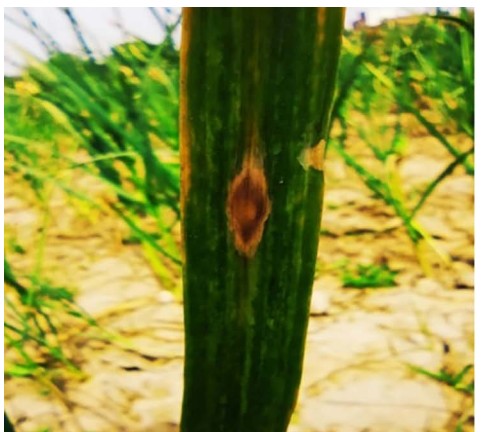 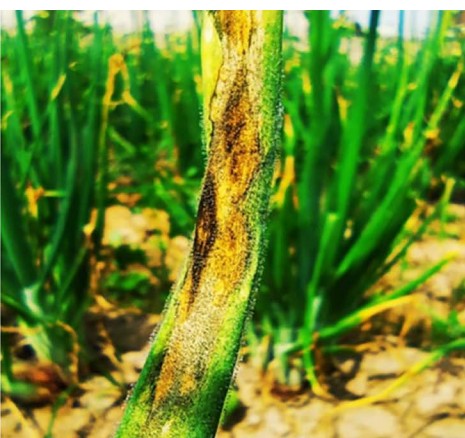

**Fig 2. Diseased onion samples showing typical symptoms of purple blotch disease.**

**Table 2. Average disease severity index after seven-day interval of post inoculation.**

| Variety | Disease Severity Index (%) | | | | |
|---|---|---|---|---|---|
| | 1st WPI | 2nd WPI | 3rd WPI | 4th WPI | 5th WPI |
| Red Imposta | 19.28[b] | 36.49[a] | 55.87[a] | 69.72[a] | 79.41[a] |
| Ceylon | 0.22[i] | 1.49[h] | 5.86[i] | 8.73[i] | 8.91[i] |
| Sultan F1 | 9.81[f] | 17.20[e] | 21.52[e] | 29.21[e] | 35.59[e] |
| Desi red | 5.57[g] | 6.78[g] | 11.87[g] | 16.64[g] | 17.13[g] |
| Mirpurkhas | 10.12[e] | 15.21[f] | 19.56[f] | 23.30[f] | 23.33[f] |
| Desi black | 23.21[a] | 29.41[c] | 42.22[c] | 61.51[b] | 67.38[b] |
| Phulkara | 0.00[j] | 0.00[i] | 3.56[j] | 6.21[j] | 8.21[j] |
| VRIO-9 | 15.51[c] | 21.56[d] | 33.58[d] | 38.21[d] | 44.53[d] |
| Early red | 2.43[h] | 5.52[g] | 9.33[h] | 11.41[h] | 13.73[h] |
| Faisalabad Red | 14.33[d] | 31.41[b] | 48.48[b] | 52.34[c] | 54.45[c] |
| Mean | 10.04 | 16.50 | 25.18 | 31.72 | 35.16 |
| P Value | <0.000 | <0.000 | <0.000 | <0.000 | <0.000 |
| LSD | 0.0782 | 1.2719 | 0.0583 | 0.1511 | 0.3131 |

Small letters with values in column showing their significance according to LSD test (P ≤ 0.05)

**Table 3. Nested Structured ANOVA for determination of Superoxide Dismutase (U/mg protein) from inoculated and un-inoculated leaves of onion plants.**

| SOV | DF | SS | MS | P | F value | Variance component | % of total variance |
|---|---|---|---|---|---|---|---|
| Type | 1 | 24626.4625 | 24626.4625 | 19.488 | 0.048* | 648.966 | 88.21 |
| Group | 2 | 2527.3751 | 1263.6876 | 13.391 | 0.017* | 64.962 | 8.83 |
| Variety | 4 | 377.4751 | 94.3688 | 7.407 | 0.000* | 9.070 | 1.23 |
| Error | 64 | 815.4221 | 12.7410 | – | – | 12.741 | 1.73 |
| Total | 71 | 28346.7348 | – | – | – | 735.39 | – |

*=Significant

**Table 4. Quantity of Superoxide Dismutase (U/mg protein) in type (inoculated and un-inoculated), reaction groups (susceptible and resistant) and varieties/advanced lines of onion plants.**

| Varieties/Advanced Lines (C) | Phulkara | | Ceylon | | VRIO-9 | | Fsd Red | |
|---|---|---|---|---|---|---|---|---|
| Group (B) | Resistant | | | | Susceptible | | | |
| Type (A) | Inocu. | Un-inocu. | Inocu. | Un-Inocu. | Inocu. | Un-Inocu. | Inocu. | Un-Inocu. |
| Amount of SOD in (C) | 33.80 | 71.37 | 37.81 | 78.52 | 24.29 | 60.80 | 28.31 | 61.48 |
| Av. amount of SOD in (C) | 52.58 | | 58.16 | | 42.54 | | 44.89 | |
| Av. amount of SOD in (B) | Resistant = 55.37 | | | | Susceptible = 43.71 | | | |
| Av. amount of SOD in (A) | Inoculated = 31.05 | | | | Un-inoculated = 68.04 | | | |

## Estimation of POD (U/mg protein)

Peroxidase (POD U/mg protein) activity in inoculated and un-inoculated onion leaves was determined using nested structured ANOVA. Statistical analysis expressed that group (susceptible and resistant) accounted for 9.53% and the varieties for 0.28% of the total variance, both showing significant differences in POD concentration (Table 5). The maximum POD

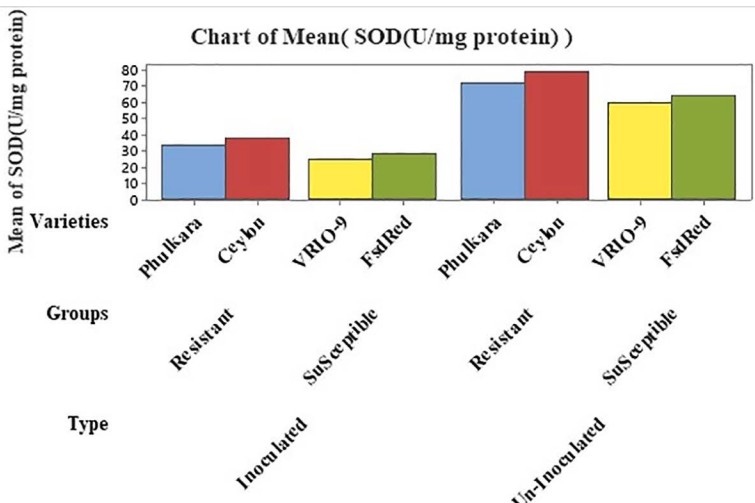

**Fig 3. SOD concentration in inoculated and un-inoculated, resistant and susceptible varieties/advanced lines of onion.**

**Table 5. Nested Structured ANOVA for determination of Peroxide (U/mg protein) from inoculated and un-inoculated leaves of onion plants.**

| SOV | DF | SS | MS | F | P | Variance Component | % of total variance |
|---|---|---|---|---|---|---|---|
| Type | 1 | 3516.0561 | 3516.0561 | 19.468 | 0.048* | 926.737 | 89.61 |
| Group | 2 | 3613.0751 | 1806.5376 | 56.716 | 0.001* | 98.594 | 9.53 |
| Variety | 4 | 127.4085 | 31.8521 | 5.346 | 0.001* | 2.877 | 0.28 |
| Error | 64 | 381.3392 | 5.9584 | – | – | 5.958 | 0.58 |
| Total | 71 | 39290.8790 | – | – | – | 1034.166 | – |

*=Significant

activity was recorded in the resistant advanced line Ceylon (43.35 U/mg protein) and Phulkara (40.40 U/mg protein). In contrast, the minimum concentration was observed in the susceptible advanced lines FSD Red (29.35 U/mg protein) and VRIO-9 (27.25 U/mg protein). A substantial difference in POD concentration was recoded between inoculated (12.98 U/mg protein) and un-inoculated plants (57.45 U/mg protein), accounting for 89.61% of total variance. Significant differences were also found between resistant (41.87 U/mg protein) and susceptible (28.3 U/mg protein) onion plants (Table 5 and 6; Fig 4).

## Estimation of CAT (U/mg protein)

Catalase (CAT U/mg protein) activity in inoculated and un-inoculated onion leaves was determined using nested structured ANOVA. The results highlighted that the group (susceptible and resistant) accounted for 10.07% and the varieties for 0.40% of the total variance, exhibited significant difference in CAT concentration (Table 7). A noticeable alteration was observed between inoculated (13.64 U/mg protein) and un-inoculated (50.87 U/mg protein) onion plants, contributing 89.36% of the total variance. Additionally, a wide difference in CAT activity was observed between resistant and susceptible groups, with concentrations of 37.92 U/mg protein and 26.59 U/mg protein, respectively. The maximum CAT activity was exhibited by the advanced resistant line Ceylon (39.16 U/mg protein), while the minimum concentration was observed in the susceptible advanced line VIRO-9 (25.23 U/mg protein) (Table 7 and 8; Fig 5).

**Table 6. Quantity of Peroxidase (U/mg protein) in type (inoculated and un-inoculated), reaction groups (susceptible and resistant) and varieties/advanced lines of onion plants.**

| Varieties/Advanced Lines (C) | Phulkara | | Ceylon | | VRIO-9 | | Fsd Red | |
|---|---|---|---|---|---|---|---|---|
| Group (B) | Resistant | | | | Susceptible | | | |
| Type (A) | Inocu. | Un-inocu. | Inocu. | Un-Inocu. | Inocu. | Un-Inocu. | Inocu. | Un-Inocu. |
| Amount of POD in (C) | 16.47 | 64.34 | 19.09 | 67.61 | 7.68 | 46.83 | 8.67 | 50.03 |
| Av. amount of POD in (C) | 40.40 | | 43.35 | | 27.25 | | 29.35 | |
| Av. amount of POD in (B) | Resistant = 41.87 | | | | Susceptible = 28.3 | | | |
| Av. amount of POD in (A) | Inoculated = 12.98 | | | | Un-inoculated = 57.45 | | | |

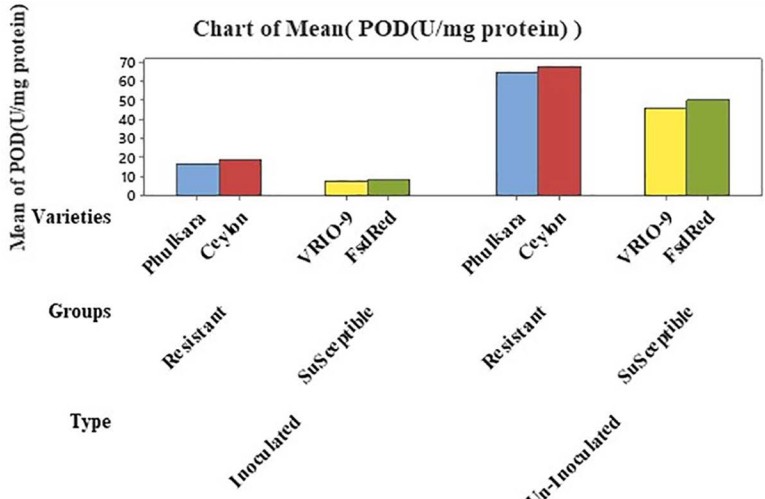

**Fig 4. POD concentration in inoculated and un-inoculated, resistant and susceptible varieties/advanced lines of onion.**

**Table 7. Nested Structured ANOVA for determination of Catalase (U/mg protein) from inoculated and un-inoculated leaves of onion plants.**

| SOV | DF | SS | MS | F | P | Variance Component | % of total variance |
|---|---|---|---|---|---|---|---|
| Type | 1 | 24457.3472 | 24457.3472 | 18.393 | 0.050* | 642.435 | 89.36 |
| Group | 2 | 2659.3891 | 1329.6946 | 49.508 | 0.002* | 72.380 | 10.07 |
| Variety | 4 | 107.4324 | 26.8581 | 20.680 | 0.000* | 2.840 | 0.40 |
| Error | 64 | 83.1198 | 1.2987 | – | – | 1.299 | 0.18 |
| Total | 71 | 27307.2886 | – | – | – | 718.953 | – |

*=Significant

## Evaluation of plant activators against purple blotch of onion under greenhouse conditions

All treatments, along with their interactions with different concentrations and time periods, expressed a significant effect against purple blotch of onion. Plants treated with Salicylic acid exhibited the least disease incidence (24.73%), followed by Citric acid (27.21%), Benzoic acid (29.32%), and $K_2HPO_4$ (33.05%) compared to the control treatment (50.13%) (Fig 6). The interaction between treatments and their concentrations indicated that salicylic acid at 0.5, 0.75

**Table 8. Quantity of Catalase (U/mg protein) in type (inoculated and un-inoculated), reaction groups (susceptible and resistant) and varieties/advanced lines of onion plants.**

| Varieties/Advanced Lines (C) | Phulkara | | Ceylon | | VRIO-9 | | Fsd Red | |
|---|---|---|---|---|---|---|---|---|
| Group (B) | Resistant | | | | Susceptible | | | |
| Type (A) | Inocu. | Un-inocu. | Inocu. | Un-Inocu. | Inocu. | Un-Inocu. | Inocu. | Un-Inocu. |
| Amount of CAT in (C) | 17.24 | 56.13 | 18.44 | 59.89 | 8.25 | 42.22 | 10.66 | 45.24 |
| Av. amount of CAT in (C) | 36.68 | | 39.16 | | 25.23 | | 27.95 | |
| Av. amount of CAT in (B) | Resistant = 37.92 | | | | Susceptible = 26.59 | | | |
| Av. amount of CAT in (A) | Inoculated = 13.64 | | | | Un-inoculated = 50.87 | | | |

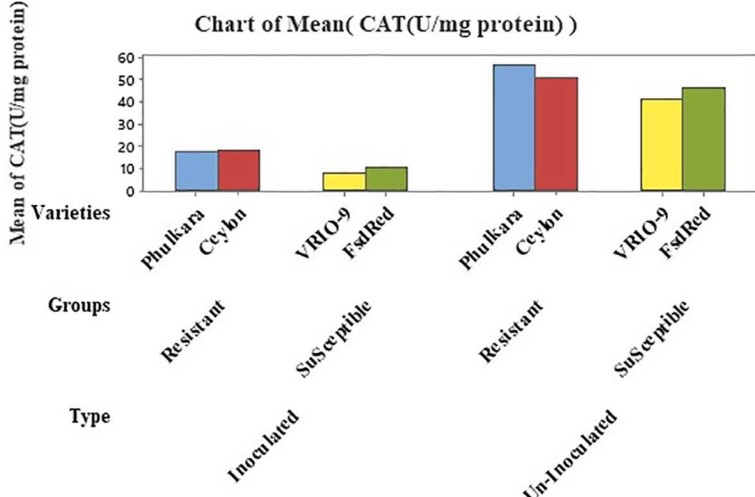

**Fig 5. CAT concentration in inoculated and un-inoculated, resistant and susceptible varieties/advanced lines of onion.**

and 1% concentrations resulted in minimum disease incidence percentage (27.2, 25.53 and 21.47%, respectively), while $K_2HPO_4$ showed the maximum incidence (35.1, 33.15 and 30.9%). Citric acid (29.33, 27.96 and 24.34%) and Benzoic acid (31.1, 29.57 and 27.3%) exhibited moderate results at the same concentrations (Fig 7). The results of the interaction between treatments and weeks indicated that, $K_2HPO_4$ was found least effective in controlling the disease, with maximum disease incidence of 36.03%, 32.61% and 30.51% after the 1st, 2nd and 3rd week of application, respectively. In contrast, Salicylic acid showed the minimum disease incidence (27.83, 24.4 and 21.97%) at the same time period (Fig 8).

### Evaluation of plant activators against purple blotch of onion under field conditions

Under open field conditions, all treatments showed significant results. The plants treated with $K_2HPO_4$ exhibited the maximum disease incidence at (46.56%), whereas salicylic acid treatment resulted in the minimum disease incidence at (32.92%), followed by Citric acid (36.91%) and benzoic acid (39%) (Fig 9). The interaction between treatments and weeks revealed that, Salicylic acid application expressed minimum disease incidence of (37.4, 33.64 and 27.73%) after 1st, 2nd and 3rd weeks of application, respectively. This was followed by citric acid (46.36, 34.66 and 29.7%) and benzoic acid (43.40, 39.4 and 35.06%). In contrast, the maximum disease incidence was exhibited by $K_2HPO_4$ with values of (50.13, 48.16 and 41.67%) at the same time intervals (Fig 10).

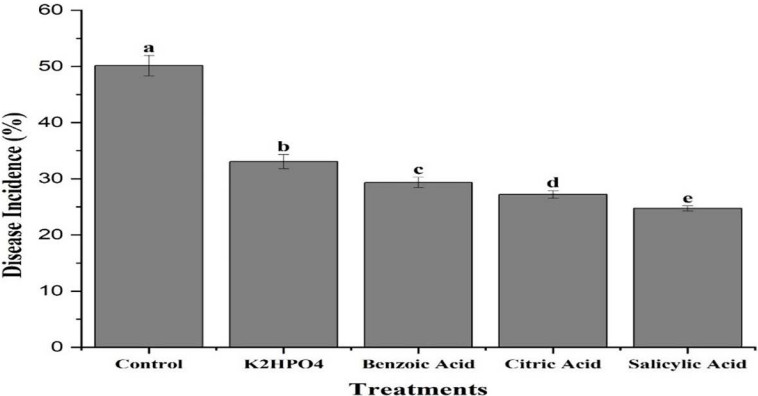

**Fig 6. Mean disease incidence of purple blotch of onion after application of plant activators under greenhouse conditions.** The bars having similar letters are not significantly different according to LSD (P ≤ 0.05).

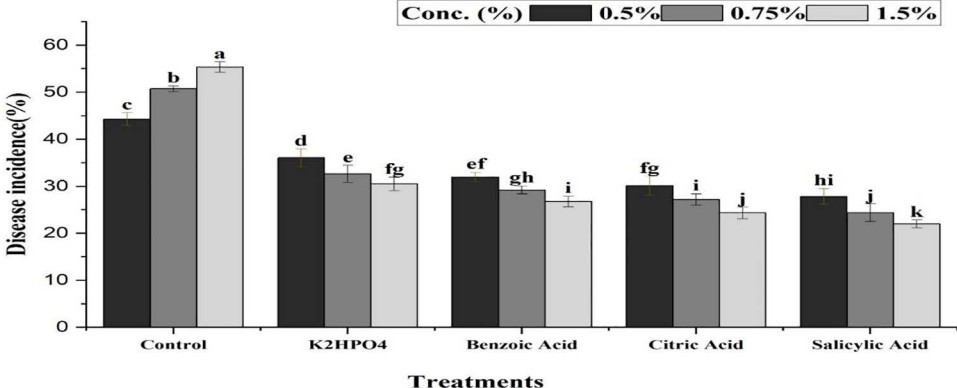

**Fig 7. Impact of interaction between plant activators and their concentrations, showing mean disease incidence of purple blotch of onion with respect to change in concentrations.** The bars having similar letters are not significantly different according to LSD (P ≤ 0.05).

## Discussion

PBD of onion, caused by the fungal pathogen *Alternaria porri,* poses a significant threat to onion production [5]. In Pakistan, fungal diseases are a major concern for successful onion cultivation. PB leads to substantial yield and economic losses worldwide [28]. The present study was conducted to isolate and identify the pathogen responsible for PBD in onion. Several studies have pointed out the role of the pathogen *A. porri* in causing PBD [5–8,20–23,29]. However, the isolated fungus was identified on the basis of its morphological characteristics, including septate mycelia and conidiophores that are geniculate, septate, straight and pale to mid-brown. This identification was carried out by following the procedure described by Woudenberg and colleagues [30]. Subsequently, a pathogenicity test was performed to confirm the association of the pathogen with the disease, where typical purplish zonate marks were observed on artificially inoculated onion plants. The findings of the pathogenicity assay are consistent with the studies of Rai and Kumari, and Thrall and coworkers[31,32]. Another study also reported signs of PBD, specifically small, depressed, oval to football-shaped lesions with a brown to purple color on infected leaves [33].

Considering the toxic effects of chemical based pesticides on human and animal health, as well as their detrimental impact on the environment, the use of resistant cultivar is the most effective and sustainable method to prevent and

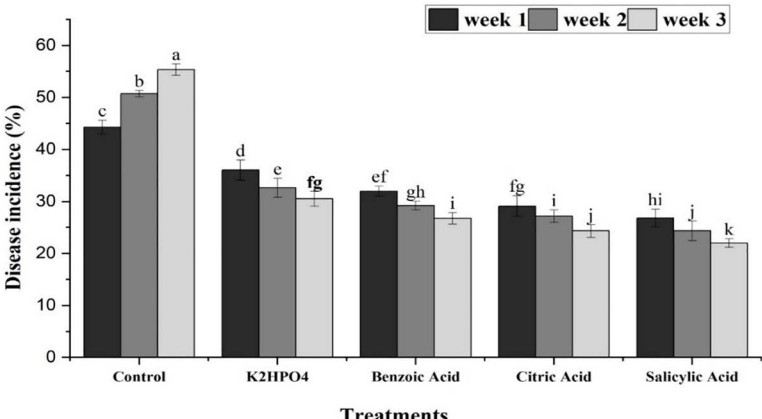

**Fig 8. Impact of interaction between plant activators and weeks, showing mean disease incidence of purple blotch of onion with respect to time intervals. The bars having similar letters are not significantly different according to LSD (P ≤ 0.05).**

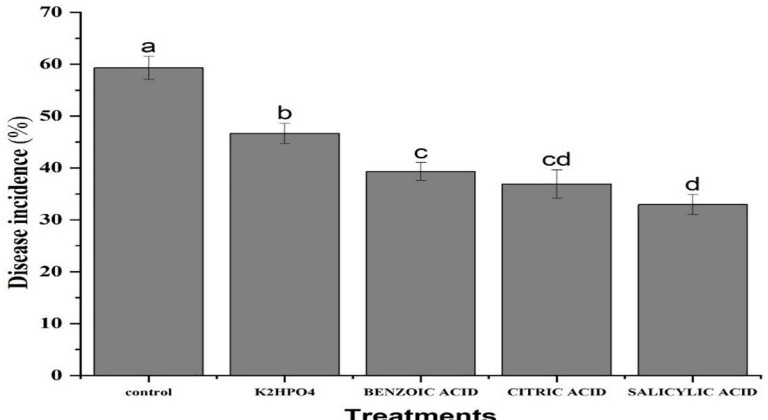

**Fig 9. Mean disease incidence of purple blotch of onion after application of plant activators under field conditions. The bars having similar letters are not significantly different according to LSD (P ≤ 0.05).**

control plant pathogenic diseases, reducing reliance on harmful chemicals [34]. Field experiments have been conducted to screen for purple blotch disease and support breeding programs for disease resistance in susceptible plants. Implementing resistant cultivars through these programs is an effective way to curb the infection, benefiting both farmers and consumers. Continued research will be essential for identifying resistance genes to enhance crop improvement [8]. The current study includes the screening of ten onion varieties/cultivars against PDB. None of the varieties were found to be immune or highly resistant; however, the results indicated that only two varieties, Phulkara and Ceylon showed a resistant response, while Desi Black and Red Imposta were highly susceptible. Our findings are consistent with those of Pathak et al. [25], where only one advance line, IR-56-1, was found to be resistant. Similarly, Alam et al. [35] reported that two advanced lines, IC49371 and IC39178, expressed resistance, while the variety Hazo showed a moderately resistant response against PBD. of onion. Another study, screening of 43 onion genotypes for purple blotch resistance under both artificial and field conditions revealed that *Allium cepa* accession 'CBT-Ac77' and the cultivar 'Arka Kalyan' were highly resistant to infection, outperforming other genotypes [36]. The specific disease reactions of different genotypes against

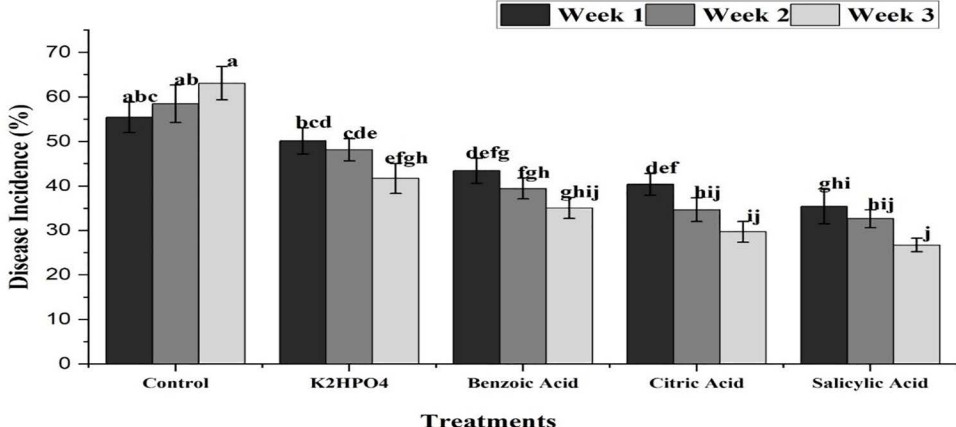

**Fig 10. Impact of interaction between plant activators and weeks, showing mean disease incidence of purple blotch of onion with respect to time intervals under field conditions. The bars having similar letters are not significantly different according to LSD (P ≤ 0.05).**

purple blotch, as evident from this study, could be highly useful for researchers in disease forecasting and integrated disease management (IDM) programs.

Enzymes involved in plant defense mechanism are crucial for helping plants combat diseases caused by pathogens. The interaction between pathogens and host plants alters the concentration of these defense enzymes, which can indicate the plants potential resistance or susceptibility to the pathogen. An excess or deficiency of these enzymes can increase the plant's vulnerability to infection [37]. In our study, we measured the activities of three key enzymes such as SOD, POD and CAT in both inoculated and un-inoculated onion cultivars, including both resistant and susceptible verities, to assess biochemical responses of PBD. Our results showed that the activities of these enzymes were maximum in un-inoculated resistant cultivars (Phulkaro and Ceylon) compared to the un-inoculated susceptible cultivars (VRIO-9 and Fsd Red). Conversely, the minimum enzymes activities were recorded in the inoculated susceptible cultivars. Our results are consistent with the findings of of Mahatma and coworkers and Nawar and Kuti [38,39], who observed similar alterations in enzymes activity in inoculated and un-inoculated resistant and susceptible plants infected with different pathogens.

Superoxide dismutase play a critical role in initiating the plants defense response. These enzymes neutralize free radicals by converting harmful reactive oxygen species into less reactive molecules. SODs are among the first enzymes produced by plants upon encountering a pathogen [40]. Meanwhile, peroxidases are responsible for the oxidation of various compounds such as organic acids, proteins, and phenols. They also enhance the mechanical strength of the plant cell wall, which supports the defense mechanism [41]. Furthermore, catalase, an essential enzyme in plant peroxisomes, help to regulate the plant immune system by breaking down hydrogen peroxide a byproduct of oxidative stress, into water and oxygen, thereby minimizing oxidative damage in plant cell [42].

Our study also investigated the management of PB in onion by using plant defense activators. Four activators including salicylic acid, citric acid, benzoic acid and $K_2HPO_4$ were used at three different concentrations (0.5, 0.75 and 1%) under greenhouse conditions. The results indicated that salicylic acid at a 1% concentration was most effective in reducing the incidence of PB caused by *A. porri*. The 1% concentration of all plant activators were further evaluated under open field conditions. Salicylic acid at this concentration recorded the lowest disease incidence percentage in the third week of application. In contrast, $K_2HPO_4$ showed the maximum disease incidence, making it the least effective against PB. Previous studies have shown that the salicylic acid (SA) induced PAL activity in onion plants, enhancing their immunity against pathogens [43]. Canaki [44] also demonstrated that SA can alter the quantity and effectiveness of pathogen-related proteins released by plants in response of pathogen attack. Our findings are consistent with those of Papoutsis [45], who

reported the potential antifungal activity of citric acid against *Alternaria alternata*. Overall, these findings support the use of resistant cultivars and salicylic acid for effective management of purple blotch, promoting more sustainable onion farming practices.

## Conclusion

Based on the results of this study, the genotypes Phulkara and Ceylon demonstrated a resistance response to PBD. These genotypes can be used for commercial production or as parents in onion variety development programs aimed at producing PB-resistant cultivars. The higher level of SOD, POD and CAT in the resistant cultivars compared to the susceptible ones suggest that biochemical profiling can help better differentiate between resistant and susceptible germ-plasm. Additionally, the present study introduces a new management approach for PB in onion using Salicylic acid, which provides systematic acquired resistance against the pathogen.

## Supporting information

**S1 Data. Data.**
(RAR)

## Acknowledgments

The authors are thankful to Vegetables' Research Area, Ayub Agriculture Research Institute (AARI), Faisalabad for providing onion seedlings and technical support and Dr. Joseph Juma Mafurah from Faculty of Agriculture, Department of Crops, Horticulture and Soils, Egerton University, Nakuru, Kenya for providing valuable suggestions and language editions for this paper.

## Author contributions

**Conceptualization:** Nasir Ahmed Rajput, Muhammad Atiq.

**Data curation:** Muhammad Usman Ali, Ghalib Ayaz Kachelo.

**Formal analysis:** Muhammad Usman Ali, Ghalib Ayaz Kachelo.

**Methodology:** Muhammad Usman, Ahmad Nawaz.

**Supervision:** Nasir Ahmed Rajput, Muhammad Atiq.

**Writing – original draft:** Muhammad Asghar.

**Writing – review & editing:** Nasir Ahmad Khan, Khalid Naveed, Owais Iqbal.

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
