## [Decision Letter · Decision Letter 0]

12 Jul 2024

PONE-D-24-22600Investigating Biochemical Variations in Onion Leaves Due to Purple Blotch Disease and its Management through Induced ResistancePLOS ONE

Dear Dr. Rajput,

Thank you for submitting your manuscript to PLOS ONE. After careful consideration, we feel that it has merit but does not fully meet PLOS ONE’s publication criteria as it currently stands. Therefore, we invite you to submit a revised version of the manuscript that addresses the points raised during the review process.

We look forward to receiving your revised manuscript.

Kind regards,

Muhammad Nauman Ahmad, PhD

Academic Editor

PLOS ONE

Journal Requirements:

3. We note that your Data Availability Statement is currently as follows: All relevant data are within the manuscript and its Supporting Information files

Reviewers' comments:

Reviewer's Responses to Questions

**Comments to the Author**

1. Is the manuscript technically sound, and do the data support the conclusions?

Reviewer #1: No

Reviewer #2: Yes

2. Has the statistical analysis been performed appropriately and rigorously? 

Reviewer #1: Yes

Reviewer #2: Yes

3. Have the authors made all data underlying the findings in their manuscript fully available?

Reviewer #1: Yes

Reviewer #2: Yes

4. Is the manuscript presented in an intelligible fashion and written in standard English?

Reviewer #1: No

Reviewer #2: Yes

5. Review Comments to the Author

Reviewer #1: This study by Asghar et al., is about onion genotypes for resistance against purple blotch and management using different plant activators.

The study is poorly planned, poorly written and the images are not clear to say it is A. porri.

Introduction:

-mention the crop loss in USD not percentages

-use more references to talk about what management options are currently available to manage A.porri and what new the study is addressing.

M&M:

-use disease severity to explain the varietal response.

-Disease control efficacy (%) = 100x[1-B/A]

Results:

Remove table 2, 3. 4

Discussion:

Use references to support/conclude the results you got.

Read more literature to see how results are presented and manuscript is written. The image quality is poor.

This work published in Plos one could be a good starting point (no conflict of interest to reviewer): https://www.ncbi.nlm.nih.gov/pmc/articles/PMC8775252/

Reviewer #2: The research article titled “Investigating Biochemical Variations in Onion Leaves Due to Purple Blotch Disease and its Management through Induced Resistance” reports the germplasm of onion against purple blotch disease, biochemical alteration analysis in onion due to pathogen invasion and evaluation of plant activators against purple blotch. This is a well-written article, and I anticipate that the manuscript should be of great interest to the researchers working on plant microbe interactions and pathogenicity. I considered the manuscript suitable for publication, subject to following improvements.

1. The abstract section should be more robust by adding some quantitative data on from the significant results.

2. When a unique noun first appears, and you want to use the abbreviation later on then the full name must be accompanied by the abbreviation. Line 33 or 77: I suggest to add “PB” in the line 33.

3. Line 41: “plant activators” add the names you evaluated.

4. Line 43: Replace “The contemporary study concluded” with “The present study revealed”.

5. Revise the statement in Line 45-46: “While salicylic acid was proved to be highly significant in controlling the purple blotch in onion.”

6. Merge paragraph No 4 from line 77-81 in the 3rd paragraph by adding a statement for coherence.

7. Line 92: remove the word available and add the names.

8. Revise the objective section of your manuscript.

9. Line 97-98: A comprehensive survey was conducted in different onion growing areas of District Faisalabad. Add prominent names of the regions you collected samples.

10. In line 104: the authors used surface sterilization for the samples and then they isolated the pathogens. It is confusing, did you isolated endophytic fungi? As you mentioned that the surface was sterilized. Justify and revise it.

11. Revise the statement Line 117-118: The identification of pathogen was conducted on the basis of morphological and microscopic characters of the isolated pathogen.

12. In results Line 217-219: How you confirmed the fungus at species level? All these phenotypic characters are common in different fungal species.

13. Cite latest references and improve the discussion section.

6. PLOS authors have the option to publish the peer review history of their article (what does this mean? ). If published, this will include your full peer review and any attached files.

**Do you want your identity to be public for this peer review?** For information about this choice, including consent withdrawal, please see our Privacy Policy .

Reviewer #1: No

Reviewer #2: **Yes: ** Sajid Ali

---

## [Author Response · Author response to Decision Letter 0]

21 Aug 2024

Point-by-point response

Reviewer #1:

This study by Asghar et al., is about onion genotypes for resistance against purple blotch and management using different plant activators. The study is poorly planned, poorly written and the images are not clear to say it is A. porri.

Response: Thank you very much for you constructive comments and suggestions. We have carefully revised the manuscript based on your feedback and made the following revisions:

-We have thoroughly reviewed and revised the manuscript to address concerns about its planning and writing. We have made significant improvements to enhance the overall clarity and consistency of the study.

-We have update the image of A. porri to improve their quality and clarity. The new image are higher resolution, and we have provided detailed figure legends to ensure accurate interpretation.

Introduction:

-mention the crop loss in USD not percentages

Response: Due to the lack of specific, accurate data on the economic losses caused by PBD worldwide, we are unable to provide precise figures. To address this, we have included a general statement in the manuscript highlighting the significant economic impact of PBD on onion crops.

-use more references to talk about what management options are currently available to manage A.porri and what new the study is addressing.

Response: We have updated the manuscript to include the references on current management option for A. porri.

M&M:

-use disease severity to explain the varietal response.

Response: In present manuscript we use tha data of disease incidence, not disease severity. It is a reason that we cannot use the disease severity in the manuscript.

-Disease control efficacy (%) = 100x[1-B/A]

Response: We cannot use the disease control efficacy formula because we use the data of disease incidence after the application of plant defense activators against PBD.

Results:

Remove table 2, 3. 4

Response: It is necessary to understand the results and design of the experiment (Nested Structured Design), so it cannot be removed.

Discussion:

-Use references to support/conclude the results you got.

Response: We have revised the manuscript to include references that support and examine our results.

-Read more literature to see how results are presented and manuscript is written. The image quality is poor.

Response: We have improved the quality of the image and manuscript by adding latest references and quality of the language.

-This work published in Plos one could be a good starting point (no conflict of interest to reviewer): https://www.ncbi.nlm.nih.gov/pmc/articles/PMC8775252/

Response: Thank you for your proposal to read out the article, you have suggested. In the light of this article, we have improved our manuscript.

Thank you again for your valuable feedback. We hope that these revisions address your concerns and improve the quality of the manuscript.

Reviewer #2:

The research article titled “Investigating Biochemical Variations in Onion Leaves Due to Purple Blotch Disease and its Management through Induced Resistance” reports the germplasm of onion against purple blotch disease, biochemical alteration analysis in onion due to pathogen invasion and evaluation of plant activators against purple blotch. This is a well-written article, and I anticipate that the manuscript should be of great interest to the researchers working on plant microbe interactions and pathogenicity. I considered the manuscript suitable for publication, subject to following improvements.

Response: Many thanks for you constructive comments and suggestions to improve this MS!

1. The abstract section should be more robust by adding some quantitative data on from the significant results.

Response: We rewrote the abstract by adding the quantitative data.

2. When a unique noun first appears, and you want to use the abbreviation later on then the full name must be accompanied by the abbreviation. Line 33 or 77: I suggest to add “PB” in the line 33.

Response: We added “PB”.

3. Line 41: “plant activators” add the names you evaluated.

Response: We added all plant activators used in this study.

4. Line 43: Replace “The contemporary study concluded” with “The present study revealed”.

Response: We replaced.

5. Revise the statement in Line 45-46: “While salicylic acid was proved to be highly significant in controlling the purple blotch in onion.”

Response: We revised the statement.

6. Merge paragraph No 4 from line 77-81 in the 3rd paragraph by adding a statement for coherence.

Response: We merged the paragraph.

7. Line 92: remove the word available and add the names.

Response: We removed.

8. Revise the objective section of your manuscript.

Response: We revised the objectives in the manuscript.

9. Line 97-98: A comprehensive survey was conducted in different onion growing areas of District Faisalabad. Add prominent names of the regions you collected samples.

Response: We added.

10. In line 104: the authors used surface sterilization for the samples and then they isolated the pathogens. It is confusing, did you isolated endophytic fungi? As you mentioned that the surface was sterilized. Justify and revise it.

Response: We used surface sterilization to ensure only internal pathogens were isolated, not surface contaminants. This method confirmed the isolated fungi were responsible for the disease symptoms, not endophytic fungi.

11. Revise the statement Line 117-118: The identification of pathogen was conducted on the basis of morphological and microscopic characters of the isolated pathogen.

Response: We revised the statement.

12. In results Line 217-219: How you confirmed the fungus at species level? All these phenotypic characters are common in different fungal species.

Response: We confirmed the fungus at the species level through detailed morphological characterization, examining distinctive traits like conidia shape, size, and septation, as well as colony morphology.

13. Cite latest references and improve the discussion section.

Response: We cited latest references and discussion section improved.

Again Many Thanks!

Thank you to both the reviewers and the editor for your valuable feedback and thoughtful contributions to improving my manuscript.

Dr. Nasir Ahmed Rajput

---

## [Decision Letter · Decision Letter 1]

8 Dec 2024

PONE-D-24-22600R1Investigating Biochemical Variations in Onion Leaves Due to Purple Blotch Disease and its Management through Induced ResistancePLOS ONE

Dear Dr. Rajput,

Thank you for submitting your manuscript to PLOS ONE. After careful consideration, we feel that it has merit but does not fully meet PLOS ONE’s publication criteria as it currently stands. Therefore, we invite you to submit a revised version of the manuscript that addresses the points raised during the review process.

We look forward to receiving your revised manuscript.

Kind regards,

Abhay K. Pandey

Academic Editor

PLOS ONE

Journal Requirements:

Reviewers' comments:

Reviewer's Responses to Questions

**Comments to the Author**

1. If the authors have adequately addressed your comments raised in a previous round of review and you feel that this manuscript is now acceptable for publication, you may indicate that here to bypass the “Comments to the Author” section, enter your conflict of interest statement in the “Confidential to Editor” section, and submit your "Accept" recommendation.

Reviewer #1: All comments have been addressed

Reviewer #2: All comments have been addressed

2. Is the manuscript technically sound, and do the data support the conclusions?

Reviewer #1: No

Reviewer #2: Yes

3. Has the statistical analysis been performed appropriately and rigorously? 

Reviewer #1: Yes

Reviewer #2: No

4. Have the authors made all data underlying the findings in their manuscript fully available?

Reviewer #1: Yes

Reviewer #2: Yes

5. Is the manuscript presented in an intelligible fashion and written in standard English?

Reviewer #1: Yes

Reviewer #2: Yes

6. Review Comments to the Author

Reviewer #1: Response: In present manuscript we use tha data of disease incidence, not disease severity. It is a reason that we cannot use the disease severity in the manuscript.

When evaluating the varieties for resistance - disease severity has to be taken into account. You have not collected the information. Your evaluation is flawed. A variety is resistant based on disease severity incidence the difference in disease severity. It is a open field experiment. Only disease incidence is not the appropriate measurement to say a variety is resistant.

Nanda et al., 2016 Identification of Novel Source of Resistance and Differential Response of Allium Genotypes to Purple Blotch Pathogen, Alternaria porri (Ellis) Ciferri (no conflict of interest to reviewer)(https://www.ncbi.nlm.nih.gov/pmc/articles/PMC5117860/pdf/ppj-32-519.pdf) - used disease incidence and disease severity to identify disease resistant variety. Expt was also conducted in invitro condition to evaluate it.

Specific comments:

Line 73,75, 231, 241, 301, 380, 448, 460 - space between words

Line 135 - specimen

Line 143 - how many days?

L161 - Is the location chosen for the expt is a hotspot for PBD?

L196-Separate

L301- recorded

Table 2&3 , Line 293, 314 - expand SOD - titles should be stand alone

Line 322 - recorded

Table 4, 5 - expand PoD

L334, Table 6&7, Figure 6 - expand CAT

Reviewer #2: The statistical analysis section is not added to the revised manuscript.

It is suggested to add a statistical analysis section in the material and methods.

7. PLOS authors have the option to publish the peer review history of their article (what does this mean? ). If published, this will include your full peer review and any attached files.

**Do you want your identity to be public for this peer review?** For information about this choice, including consent withdrawal, please see our Privacy Policy .

Reviewer #1: No

Reviewer #2: **Yes: ** Sajid Ali

---

## [Author Response · Author response to Decision Letter 1]

17 Mar 2025

Point-by-point response

Reviewer #1:

When evaluating the varieties for resistance - disease severity has to be taken into account. You have not collected the information. Your evaluation is flawed. A variety is resistant based on disease severity incidence the difference in disease severity. It is a open field experiment. Only disease incidence is not the appropriate measurement to say a variety is resistant.

Nanda et al., 2016 Identification of Novel Source of Resistance and Differential Response of Allium Genotypes to Purple Blotch Pathogen, Alternaria porri (Ellis) Ciferri (no conflict of interest to reviewer)(https://www.ncbi.nlm.nih.gov/pmc/articles/PMC5117860/pdf/ppj-32-519.pdf) - used disease incidence and disease severity to identify disease resistant variety. Expt was also conducted in invitro condition to evaluate it.

Response: Thank you very much for your suggestions and improvements. We have re-conducted our experiment after obtaining additional time from the editor for revision. As per your suggestion, we have added new data on disease severity instead of disease incidence.

Specific comments:

1). Line 73,75, 231, 241, 301, 380, 448, 460 - space between words

Response: All typo mistakes has been acknowledged.

2). Line 135 – specimen

Response: Corrected

3). Line 143 - how many days?

Response: Added as “The fungal pieces were transferred to new sterilized petri plates containing fresh potato dextrose agar (PDA) and incubated at 25°±5°C for seven days.”

4). L161 - Is the location chosen for the expt is a hotspot for PBD?

Response: Yes, PBD was frequently observed at the location (Vegetables' Research Area, Ayub Agriculture Research Institute (AARI), Faisalabad) of experiments

5). L196-Separate

Response: Corrected

6). L301- recorded

Response: Corrected

7). Table 2&3 , Line 293, 314 - expand SOD - titles should be stand alone

Response: Corrections has been incorporated

8). Line 322 – recorded

Response: corrected

9). Table 4, 5 - expand PoD

Response: We have added full names.

10). L334, Table 6&7, Figure 6 - expand CAT

Response: Acknowledged in tables and fig.

Reviewer #2:

1). The statistical analysis section is not added to the revised manuscript.

It is suggested to add a statistical analysis section in the material and methods.

Response: The data related to statistical analysis in methodology section has been added.

Thank you to both the reviewers and the editor for your valuable feedback and thoughtful contributions to improving my manuscript.

---

## [Editor Report · Decision Letter 2]

2 Apr 2025

Investigating Biochemical Variations in Onion Leaves Due to Purple Blotch Disease and its Management through Induced Resistance

PONE-D-24-22600R2

Dear Dr. Rajput,

We’re pleased to inform you that your manuscript has been judged scientifically suitable for publication and will be formally accepted for publication once it meets all outstanding technical requirements.

Kind regards,

Abhay K. Pandey

Academic Editor

PLOS ONE

Additional Editor Comments (optional):

authors addressed all reviewer comments
---

## [Editor Report · Acceptance letter]

PONE-D-24-22600R2

PLOS ONE

Dear Dr. Rajput,

I'm pleased to inform you that your manuscript has been deemed suitable for publication in PLOS ONE. Congratulations! Your manuscript is now being handed over to our production team.

Kind regards,

on behalf of

Dr. Abhay K. Pandey

Academic Editor

PLOS ONE